# Determinants of Dutch public health professionals' intention to use digital contact tracing support tools: A cross-sectional online questionnaire study

Yannick B. Helms[1]*, Mart L. Stein[1], Nora Hamdiui[1], Akke van der Meer[1], José A. Ferreira[2], Rik Crutzen[3], Aura Timen[1,4], Mirjam E. E. Kretzschmar[1,5]

1 Centre for Infectious Disease Control (CIb), National Institute for Public Health and the Environment (RIVM), Bilthoven, The Netherlands, 2 Department of Statistics, Informatics and Modelling, National Institute for Public Health and the Environment (RIVM), Bilthoven, The Netherlands, 3 Department of Health Promotion, Care and Public Health Research Institute (CAPHRI), Maastricht University, Maastricht, The Netherlands, 4 Department of Primary and Community Care, Radboud Institute for Health Sciences, Radboud University Medical Center, Nijmegen, The Netherlands, 5 Julius Center for Health Sciences and Primary Care, University Medical Center Utrecht, Utrecht University, Utrecht, The Netherlands

* yannick.helms@rivm.nl

**Data Availability Statement:** All relevant data are within the manuscript and its Supporting Information files.

## Abstract

Contact tracing (CT) can be a resource intensive task for public health services. To alleviate their workload and potentially accelerate the CT-process, public health professionals (PHPs) may transfer some tasks in the identification, notification, and monitoring of contacts to cases and their contacts themselves, using 'digital contact tracing support tools' (DCTS-tools). In this study, we aimed to identify determinants of PHPs' intention to use DCTS-tools. Between February and April 2022, we performed a cross-sectional online questionnaire study among PHPs involved in CT for COVID-19 in the Netherlands. We built three random forest models to identify determinants of PHPs' intention to use DCTS-tools for the identification, notification, and monitoring of contacts, respectively. The online questionnaire was completed by 641 PHPs. Most respondents had a positive intention towards using DCTS-tools for the identification (64.5%), notification (58%), and monitoring (55.2%) of contacts. Random forest models were able to correctly predict the intention of 81%, 80%, and 81% of respondents to use DCTS-tools for the identification, notification, and monitoring of contacts, respectively. Top-determinants of having a positive intention are the anticipated effect of DCTS-tools on the feasibility and efficiency of CT (speed, workload, difficulty), the degree to which PHPs anticipated that cases and contacts may find it pleasant and may be willing to participate in CT using DCTS-tools, and the degree to which PHPs anticipated that cases and contacts are sufficiently supported in CT when using DCTS-tools. Most PHPs have a positive intention to involve cases and their contacts in the identification, notification, and monitoring stages of the CT-process through DCTS-tools. The identified top-determinants should be prioritized in the (future) development and implementation of DCTS-tools in public health practice. Citizens' perspectives on the use of DCTS-tools should be investigated in future research.

**Funding:** We acknowledge the European Commission's Horizon 2020 research and innovation program for its funding under grant agreement No. 101003480 (https://www.coresma. eu). The funders had no role in study design, data collection and analysis, decision to publish, or preparation of the manuscript. Authors who received the award: M.L.S., A.T., M.E.E.K.

**Competing interests:** The authors have declared that no competing interests exist.

## Author summary

Contact tracing can reduce the spread of diseases such as COVID-19. However, during large outbreaks public health services may lack the resources to effectively perform contact tracing. In this study, we investigated if this may be addressed by shifting some tasks that are normally performed by public health professionals to patients (cases) and their contacts, using digital tools. We surveyed 641 public health professionals involved in COVID-19 contact tracing in the Netherlands. We found that most professionals were positive about more actively involving cases and their contacts in tasks like identifying, notifying, and monitoring contacts. The belief that this approach would make contact tracing more feasible and efficient, and the belief that cases and contacts may be more willing to participate in contact tracing when given more autonomy, were important considerations for professionals. Our findings provide useful insights for the development and implementation of new approaches to contact tracing during future outbreaks. Future research should focus on understanding the views and needs of citizens regarding more actively participating in contact tracing, using digital tools.

## Introduction

Contact tracing (CT) is an important tool for containing outbreaks of communicable diseases that spread via close physical proximity or (in)direct physical contact between individuals, such as SARS-CoV-2 (COVID-19), tuberculosis, and measles. CT typically starts with interviewing *cases* (individuals with a newly confirmed infection with a given communicable pathogen) to identify *contacts* (individuals who are at risk of infection because they were in close physical proximity to the case). The contacts are then notified about their exposure and informed about what *CT-measures* (e.g., testing, quarantine, and/or isolation) may be required to prevent further spread of a pathogen. Finally, contacts' health and their implementation of —and adherence to—CT-measures are monitored and advised upon during their incubation and/or infectious period [1,2].

Previous evidence shows that CT can reduce the transmission and mortality of communicable diseases, if it is performed in a timely and sufficiently complete manner [3,4]. However, CT is a time-consuming endeavor that requires a substantial amount of labor from public health professionals (PHPs). Therefore, public health services (PHS), who are typically responsible for CT in practice, may have insufficient human resources at their disposal to perform CT, especially during pandemics or local outbreaks [5]. This can lead to delays in CT and/or may force PHS to scale-down CT (e.g., in the sense that PHS increasingly focus their resources on high-risk individuals and settings), both of which can reduce the effectiveness of CT [6]. Therefore, it is important that novel solutions to accelerate CT and to reduce the human resource needs of PHS for CT are developed and implemented.

One potential solution may be to shift some tasks in the execution of CT from PHPs to cases and contacts, using *digital contact tracing support* (DCTS) *tools*. By DCTS-tools we mean digital tools which cases and their contacts can use to support the 'traditional' (sometimes also referred to as 'manual') execution of CT by PHPs, rather than proximity tracing applications that use Bluetooth and/or Global Positioning System (GPS). For example, one study conducted in San Francisco (California, United States) during the COVID-19 pandemic evaluated the use of a chatbot to inform (relatively low-risk) cases about CT-guidelines and -measures, assess their ability to isolate, and elicit at-risk contacts. This study found that, with help from the

chatbot, PHS were able to reduce the proportion of cases assigned to a telephone interview by 31.5%, allowing PHPs to direct more resources towards relatively vulnerable communities [7]. Another study conducted in the United States investigated the perspectives of individuals who recently tested for COVID-19 on using website services for (anonymous) contact notification. This study found that 40.3% of their respondents would be interested in using such a service. In addition, respondents generally anticipated that they would notify more contacts—and notify contacts faster—if the service would be available [8]. Several studies also investigated the application of self-monitoring tools for cases and/or contacts. Overall, these studies found that digital self-monitoring tools in the form of, for example, websites or two-way text messaging services can decrease the workload for PHPs and increase the number of cases/contacts from whom health data can be collected [9–12].

More extensive research on DCTS-tools has been conducted in the context of sexually transmitted infections (STIs). Although CT for STIs and CT for close-contact pathogens can differ in many ways (e.g., in terms of exposure-risk assessment guidelines and sources of shame and stigma that can influence the CT-process), these studies yielded similar results. For example, various studies on digital partner notification services (i.e., patient-led contact notification in the context of STIs) using e-mail, SMS, and/or mobile applications, found that such services can reduce the workload of PHPs, accelerate the CT-process, and increase the number of sexual contacts notified [13–18].

Although these studies indicate that DCTS-tools indeed have the potential to benefit public health practice, they typically only focus on a small part of the CT-process and only limitedly investigate the overarching concept of more actively involving cases and contacts in CT through digital tools. In addition, the intention, perspectives and needs of PHPs regarding the use of DCTS-tools are rarely investigated. This is problematic, since DCTS-tools ultimately need to be embedded in CT-systems and -protocols, which are initially implemented by PHPs. Understanding PHPs' intention to use DCTS-tools (and the determinants thereof) is therefore important to guide the development, implementation, and use of DCTS-tools in practice.

We, therefore, previously conducted two studies among Dutch PHPs [19,20]. In the first study, we used interviews (N = 12) and online questionnaires (N = 70) to explore the opportunities and challenges of digitally involving cases and their contacts in CT under different circumstances (i.e., in the context of various close-contact pathogens and outbreak scenarios). In the second study, we performed interviews (N = 17) focused specifically on exploring PHPs' perspectives and needs regarding the application of three types of DCTS-tools in the context of the SARS-CoV-2 (COVID-19) pandemic, one for each stage of the CT-process (i.e., contact identification, notification, and monitoring). With *DCTS-tool 1*, cases may be asked to identify their contacts, collect their contacts' data, and share these data with PHS. With *DCTS-tool 2*, cases may be asked to notify (a selection of) their contacts of their exposure-risk and of the CT-measures that are required to prevent further spread of the pathogen. With *DCTS-tool 3*, contacts may be asked to self-monitor and register their symptoms and health status. We found that Dutch PHPs are generally open towards involving cases and contacts in CT through these types of DCTS-tools, as they anticipated that this approach could make CT more feasible and efficient, give cases and contacts more opportunities to participate in CT in a manner that (better) suits them, and improve the collection of CT-data (e.g., for surveillance purposes). Nevertheless, PHPs were concerned about whether CT would be adequately executed when it is relatively dependent on the willingness and skills of cases and contacts, and whether cases and contacts would receive sufficient (emotional and practical) support in the CT-process. We also identified several needs of PHPs regarding the development and implementation of DCTS-tools. These included having the possibility of automatically transferring the data provided by cases and contacts to the case-management software used at PHS, taking into account

how willing cases and contacts are to participate in CT with DCTS-tools and how skillful they are in the use of DCTS-tools (e.g., by also providing alternative ways to participate in CT, tailored to individuals' needs), and allowing PHPs to support and guide cases and contacts in the CT-process (e.g., through a chat functionality).

While our previous work has provided insight into the range of factors that influence PHPs' intention to use DCTS-tools, our understanding of these factors is still limited in the sense that we have not yet established 'what matters most' in a larger population of PHPs. As such, it is unclear what the drivers (i.e., determinants) of PHPs' intention are that should be prioritized in the development and implementation of DCTS-tools. We, therefore, performed an online questionnaire study among PHPs involved in the execution of CT for COVID-19 in the Netherlands, to improve our understanding of the determinants of PHPs' intention to use DCTS-tools. The research question that guided this study was: '*What are the determinants of Dutch PHPs' intention to involve cases and their contacts in the identification, notification, and monitoring stages of the COVID-19 CT-process, using DCTS-tools*?'

## Material and methods

### Study design

We performed a cross-sectional online questionnaire study. We followed the STROBE checklist for cross-sectional studies for reporting on the implementation and results of the study (see S1 Appendix).

### Setting and study population

CT in the Netherlands is usually performed at PHS, by specialized nurses under supervision of physicians. There are 25 PHS in the Netherlands, each representing a different geographical region.

This study was performed between February and April 2022. At that time, the number of newly identified COVID-19 cases per day was particularly high in the Netherlands, but regulations concerning contact restrictions to reduce transmission of COVID-19 (e.g., social distancing, maximum group sizes) were gradually being lifted due to increasing vaccination rates, diminishing hospitalizations rates of COVID-19 patients and the emergence of less severe variants of the virus. CT was still implemented on a large scale, but mostly in a (very) scaled-down manner. Therefore, PHS had temporarily hired many new contact tracers (e.g., medical students) and focused the responsibilities of nurses and physicians towards relatively complex clusters and the overall coordination of CT. Although national guidelines for CT were in place, the precise execution of CT (e.g., the degree to which CT was scaled down) often differed between PHS, depending on regional caseload and available personnel for CT.

To minimize bias related to different CT practices between PHS and/or PHPs with different professional backgrounds and experience with CT, we included public health nurses and physicians, and temporary contact tracers (all of whom we refer to as PHPs) from all PHS in the Netherlands in our study. Eligibility criteria were: (1) being 18 years or older, (2) being or having been involved in the execution, coordination, and/or organization of CT for COVID-19 and/or other communicable diseases, and (3) currently working for a PHS in the Netherlands.

### Sampling

We distributed the online questionnaire through two main routes, using a combination between convenience and purposive sampling. First, using the professional network of the Dutch Coordination Centre for Communicable Disease Control (LCI) of the Dutch National

Institute for Public Health and the Environment (named the RIVM), we sent an invitation by e-mail to the 25 PHS in the Netherlands with information about the study and a link to the online questionnaire. We asked the PHS to distribute the e-mail further among their public health nurses and physicians involved in the execution of CT for COVID-19. Second, we asked the umbrella-organization of PHS in the Netherlands (named GGD GHOR Nederland) to distribute an invitation by e-mail to the coordinator of temporary contact tracers in each PHS. The coordinators were asked to distribute the online questionnaire further among the temporary contact tracers working at their respective PHS. In addition, we asked all individuals who participated in the questionnaire to forward the invitation to their colleagues ('snowball sampling'). Two weeks later, we distributed the online questionnaire again, in the same manner and through the same channels, as a reminder.

Respondents who clicked on the participation link were first directed to a webpage where they were more extensively informed about the study and asked for their consent to participate. Respondents who agreed to the terms and conditions of the study and clicked 'Continue to the questionnaire' were redirected to the questionnaire.

The questionnaire was implemented and distributed using 'respondent-driven sampling' (RDS) software, developed by the Karolinska Institute, the University Medical Centre Utrecht, and the RIVM. The RDS-software is compliant with general data protection regulation guidelines and the latest information security standards [21].

## Online questionnaire

We developed and piloted the online questionnaire with four PHPs (one public health nurse, one public health physician, and two temporary contact tracers) involved in the execution of CT for COVID-19 at different PHS. Questionnaire items were mainly constructed based on the findings of previous research in which we used the Innovation Decision-Process model [22] and the Reasoned Action Approach [23] as theories to explore PHPs' perspectives and elicit their beliefs regarding the use of DCTS-tools [19,20]. As such, all questionnaire items reflected potential determinants of PHPs' intention to use DCTS-tools in practice. The questionnaire (translated from Dutch to English) can be found in S2 Appendix.

The questionnaire consisted of five main sections and took approximately 20 minutes to complete. *Section 1* (6 questions) focused on respondents' socio-demographic characteristics and professional background, namely age, gender, province of employment, primary occupation at PHS (nurse, physician, temporary contact tracer, other), experience (in years) with CT for COVID-19, and experience (yes/no) with CT for communicable diseases other than COVID-19. *Section 2* (8 questions) focused on respondents' general beliefs regarding CT for COVID-19 (e.g., workload, usefulness, effectiveness). *Section 3* (8 questions) focused on respondents' beliefs regarding the roles and responsibilities of cases, contacts, and the PHS in CT for COVID-19. *Section 4* (10 questions) focused on respondents' beliefs regarding digitalization of the CT-process. In *section 5*, we introduced and questioned respondents about DCTS-tools 1, 2, and 3 (13, 16, and 12 questions, respectively), using illustrations designed specifically for this purpose. See Figs 1–3.

The DCTS-tools depicted in Figs 1–3 were conceptualized in line with the perceptions and needs of PHPs that we identified in our previous research [19,20]. For each DCTS-tool separately, we first showed respondents the corresponding Illustration and instructed them to consider a situation in which they would be able to use the given DCTS-tool, compared to a situation in which they would perform CT 'as usual' (i.e., without the DCTS-tool). In addition, we instructed respondents to consider situations in CT in which 'standard' CT-protocols apply, rather than those in which more specialized CT-protocols apply, for example during

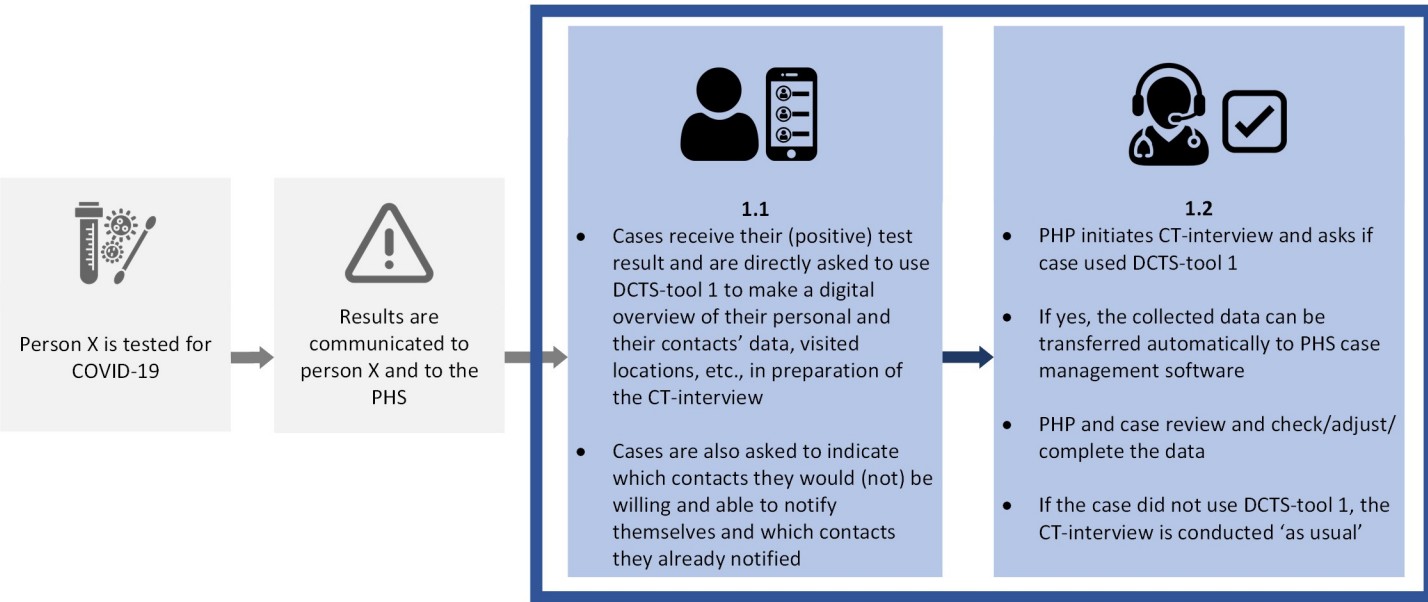

**Fig 1. Cases collect their personal and their contacts' data using DCTS-tool 1.** Grey boxes refer to moments in the CT-process; blue boxes describe the use of DCTS-tool 1.

outbreak in care-facilities, schools, or other relatively complex/impactful situations. Respondents were then questioned about their beliefs regarding benefits and challenges (e.g., regarding the speed of CT and the workload it requires/involves) and their intention to use the introduced DCTS-tool.

All questions in sections 2 to 5 were asked on 5-point Likert-scales (with values ranging from 0 to 4). Where possible, scales were formulated to measure two ends of a given spectrum.

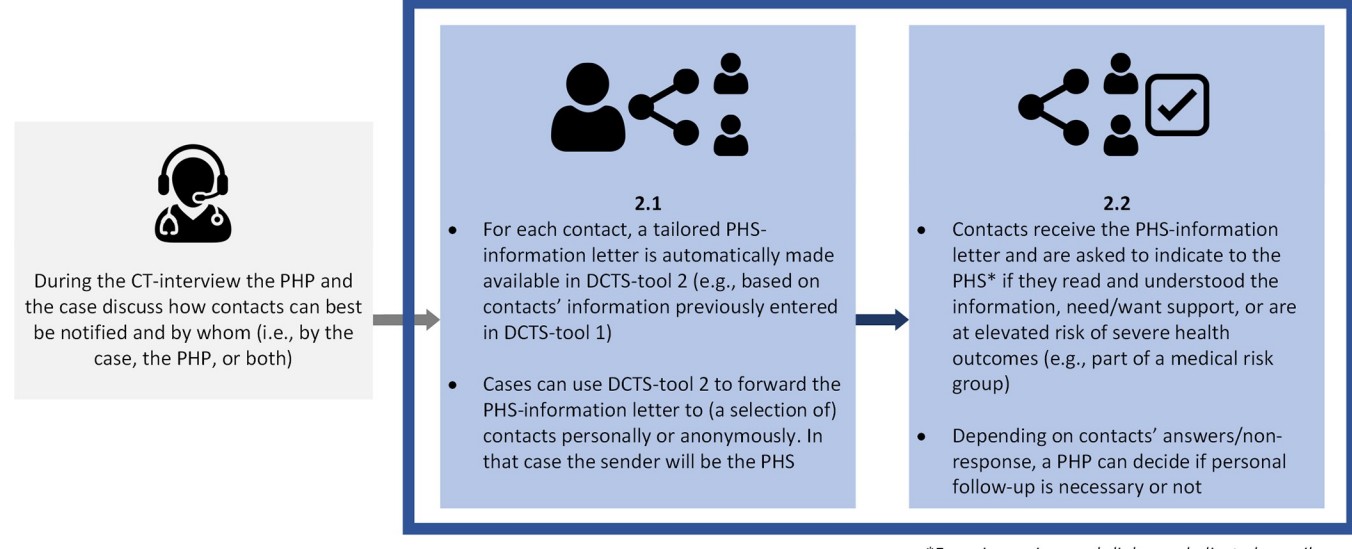

**Fig 2. Cases digitally inform (a selection of) their contacts themselves using DCTS-tool 2.** Grey boxes refer to moments in the CT-process; blue boxes describe the use of DCTS-tool 2.

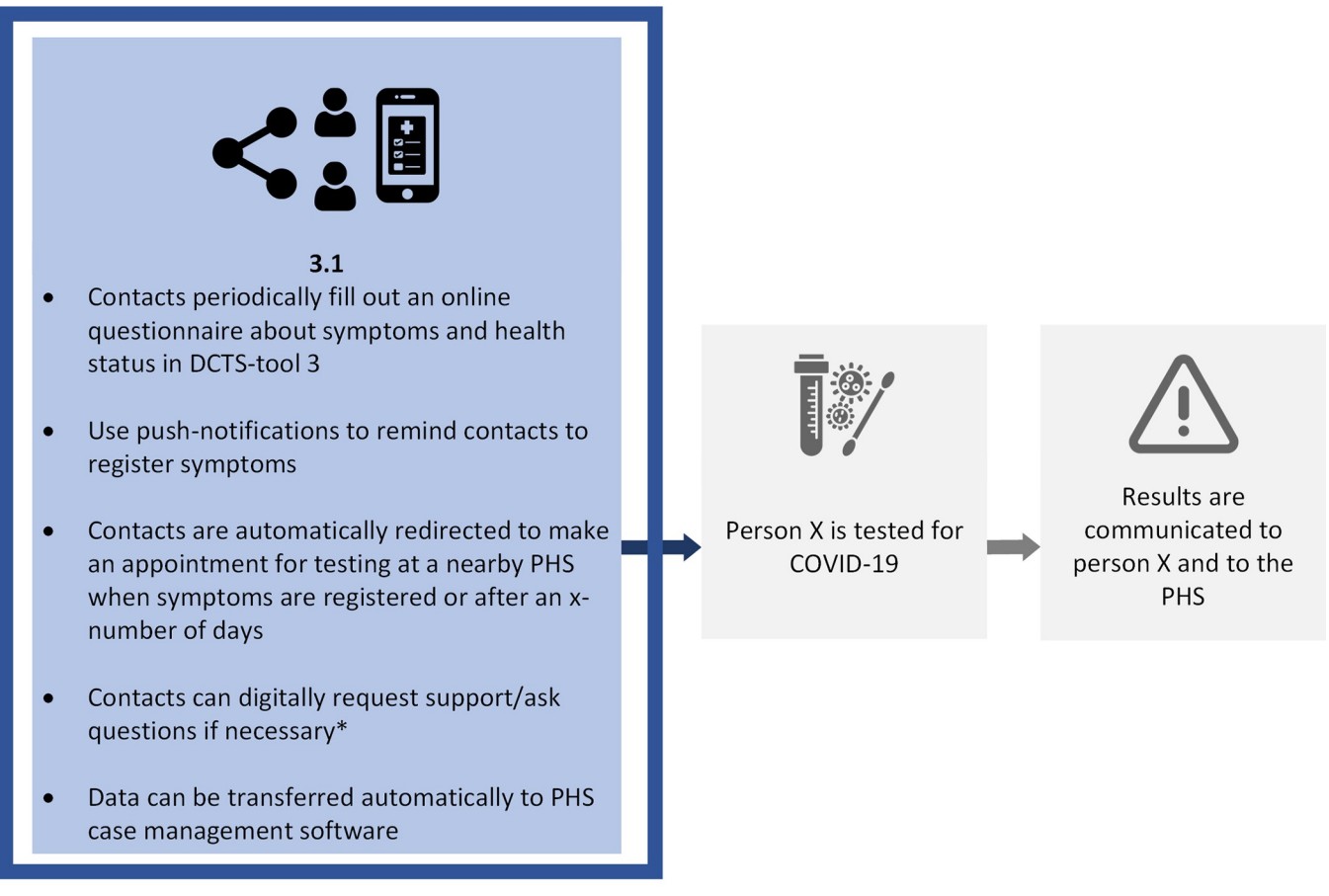

*E.g., via a unique web-link or a dedicated e-mail address or phone number*

**Fig 3. Contacts self-monitor and digitally register symptoms using DCTS-tool 3.** Grey boxes refer to moments in the CT-process; blue boxes describe the use of DCTS-tool 3.

For example, questions about the anticipated effect of DCTS-tools on the speed of CT had an answer scale ranging from 'much slower' to 'much faster' and questions regarding the anticipated willingness of cases/contacts to participate in CT had an answer scale ranging from 'very unwilling' to 'very willing'. The value '0' usually represented the most negative/undesirable side of the scale, '2' represented the middle of the scale (i.e., neutral, no effect, or not different from regular CT-practices), and '4' the most positive/desirable side of the scale.

## Statistical analyses

**Preparation of the dataset.** In our analyses, we only considered respondents who completed the online questionnaire, meaning they provided answers to all questions. Respondents who did not complete the questionnaire were discarded. Unnecessary columns, such as respondent ID's and the dates and times of starting and finishing the questionnaire were also discarded. No further data cleaning was necessary/performed.

**Descriptive statistics.** Descriptive analyses based on the sample of respondents who completed the questionnaire were performed on all variables. Distributions were investigated using percentages for categorical variables, and means (M), standard deviations (SD), medians, and inter-quartile ranges for ordinal/interval variables. In the main text, we only report

percentages, or M and SD (M; SD). See Tables A and B in S3 Appendix for a more detailed overview of our descriptive analyses.

**Bivariate analyses.** Bivariate associations between each variable and respondents' intention to use DCTS-tools 1, 2, and 3, were tested by the Mann-Whitney or Kruskal Wallis tests when one of the variables was categorical and by Spearman's test when both variables were ordinal/interval. Results of the bivariate analyses are reported in Table B in S3 Appendix.

**Multivariable analyses: Identifying determinants of PHPs' intention to use DCTS-tools.**
Using all questionnaire items as predictors, we carried out *random forest* (RF) analyses to identify determinants of PHPs' intention to use DCTS-tools 1, 2, and 3, respectively, in CT for COVID-19 [24]. RF is a non-parametric machine learning algorithm which predicts an outcome based on a set of variables. RF yields a variable importance ranking (VIR) that reflects the relative contribution, or 'importance', of each variable to the accuracy of the predictions. The importance of a variable represents the increase in prediction error, in this study measured in terms of the probability of misclassification (PMC), resulting from the replacement of the variable's value by a randomly chosen value drawn from the variable's distribution. Thus, the greater the increase in the prediction error due to this 'corruption', the greater the importance of the variable.

Using the sample of respondents who completed the questionnaire, we trained three RF-models (one for each DCTS-tool). In each model, we used a dichotomized intention variable as the outcome. Respondents who stated that they would 'definitely' or 'probably' want to use DCTS-tool 1, 2, or 3 in CT for COVID-19 were classified as having a 'positive' intention to use the respective DCTS-tool. Respondents who stated that they would 'maybe', 'probably not', or 'definitely not' want to use a particular DCTS-tool in CT for COVID-19 were classified as having a 'neutral/negative' intention. In each model, we included all general questionnaire items (sections 1–4) and questionnaire items pertaining specifically to the anticipated advantages of DCTS-tool 1, 2, or 3 (section 5) as predictors.

For each model, we report the VIR. Note that in each VIR, the value of 'Increase in pmc' on the x-axis reflects the model's absolute increase in PMC when a given variable is 'corrupted', as previously explained. We assessed the VIRs visually to identify the most important predictors (i.e., determinants of PHPs' intention) for each model. As a general guideline, we considered all predictors above the 'cut' from where predictors start to align vertically to the left side of the VIR to be determinants of PHPs' intention (predictors below this 'cut' have little contribution to a model's performance).

Each model's performance was measured by estimates of the PMC, sensitivity (SENS, the probability of a correct prediction among 'positive' individuals), and specificity (SPEC, the probability of a correct prediction among 'neutral/negative' individuals). In addition, we determined the area under the receiver operating characteristic curve (AUC) for each model. The AUC is a commonly used metric to quantify the performance of classification models. Its value ranges between 0 and 1, with 0.5 corresponding to 'random guessing' and 1 to perfect guessing. As a general guideline, an AUC-value of 0.9–0.99 can be considered as 'excellent', 0.8–0.89 as 'good', 0.7–0.79 as 'fair', and 0.51–0.69 as 'poor' [25].

Instead of logistic or linear regression, we chose RF because of its flexibility in dealing with many variables (around 45 per model in our case) and its inherent greater ability to mimic the behavior of the data when based on a large sample [26]. In addition, since we derived our questionnaire items from previously identified qualitative themes (meaning that many questionnaire items are conceptually related), we expected the predictors to be substantially correlated, which is problematic in regression analyses [27]. See also Figs A, B, C, D, E and F in S3 Appendix, where we examined this in more detail using correlation matrices and agglomerative hierarchical cluster analyses [28]. RF is insensitive to multicollinearity and it describes the joint working of the predictors in a realistic way.

**Understanding the relationship between determinants and PHPs' intention in more detail.** To investigate the relationship between the identified determinants and PHPs' intention in more detail, we constructed partial dependence plots [29]. Each plot refers to a given determinant and shows estimates of the average model-based probability of a positive intention towards a given DCTS-tool in CT for COVID-19 as a function of the value that the determinant in question is conditioned to take. As such, the plots give an idea about the direction of the relationship between a given determinant and PHPs' intention, which we may label as positive (+) or negative (-). In addition, a steeper probability curve indicates a greater effect of the determinant in question. Therefore, we also report the difference (%) between the lowest and the highest marginal probability (MP) for each determinant. For example, an MP difference of 20% indicates that the average model-based probability of a positive intention to use a DCTS-tool can differ by up to 20%, depending on the given determinant. Thus, the higher the MP difference, the larger the effect of a given determinant on the outcome. For each determinant, we report the lowest and the highest MP (MP lowest–highest), the difference (%) between them, and the direction of the relationship as positive (+) or negative (-).

All statistical analyses were performed in RStudio Server (V. 2023.03.0). We used the 'randomForest' package to perform the RF-analyses and the 'DALEX' package to calculate MPs and construct the partial dependence plots [24,30].

**Assessment of bias due to respondent dropout.** Since we only included individuals who completed the online questionnaire in our analyses, the results of this study may be biased if, for example, questionnaire completion was associated with certain demographic characteristics and/or with having a positive or negative intention towards the use of DCTS-tools. Therefore, to assess if respondent dropout may have biased our study results, we included a variable measuring questionnaire completion (yes/no) in the full dataset (also including individuals who did not complete the questionnaire) and computed bivariate associations between questionnaire completion and respondents' demographics using chi-square and t-tests. In addition, we rebuilt the first RF-model (for DCTS-tool 1) with the full dataset, including the questionnaire completion variable as a predictor.

Although some demographic characteristics were statistically significantly associated with questionnaire completion, we did not find that the differences between the groups of individuals who completed the questionnaire and those who did not were meaningful/large. The additional RF-analysis indicated that questionnaire completion had a negligible effect on PHPs' intention. This suggests that respondent dropout may have only limitedly influenced our results and findings. See also Table C and Fig G in S3 Appendix.

## Ethical considerations

This study was reviewed by the Medical Ethical Review Committee of the University Medical Centre Utrecht, who exempted this study from the need for a full medical ethical review (reference number: 21-715/C).

## Results

### Sample characteristics

Eight hundred and sixty-two individuals consented to participate in the study and started the online questionnaire, of whom 641 completed all questions and were included in the analyses (74% completion rate). None of the respondents was excluded for not meeting the eligibility criteria.

The mean age of respondents was 40.7 (SD = 15.4) years. Respondents were employed at PHS from all provinces in the Netherlands. The primary occupation of 11.1% of respondents

**Table 1. Sample characteristics.**

| Characteristic | Individuals who completed the online questionnaire (N = 641) |
|---|---|
| **Age** (Mean (SD); Med (IQR)) | Mean (SD) = 40.7 (15.4)<br>Med (IQR) = 38 (28–53) |
| **Gender** (%) | |
| • Female | 462 (72.1%) |
| • Male | 177 (27.6%) |
| • Non-binary | 2 (0.3%) |
| **Province of employment** (%) | |
| • Drenthe | 4 (0.6%) |
| • Flevoland | 15 (2.3%) |
| • Friesland | 71 (11.1%) |
| • Gelderland | 143 (22.3%) |
| • Groningen | 27 (4.2%) |
| • Limburg | 58 (9.0%) |
| • Noord-Brabant | 37 (5.8%) |
| • Noord-Holland | 70 (10.9%) |
| • Overijssel | 70 (10.9%) |
| • Utrecht | 57 (8.9%) |
| • Zeeland | 34 (5.3%) |
| • Zuid-Holland | 50 (7.8%) |
| • Working at PHS in multiple provinces | 5 (0.8%) |
| **Primary occupation at PHS** (%) | |
| • CT-manager/coordinator | 25 (3.9%) |
| • Temporary contact tracer | 538 (83.9%) |
| • PHS-physician | 39 (6.1%) |
| • PHS-nurse | 32 (5.0%) |
| • Other (conversation coach, health educator, policy advisor) | 7 (1.1%) |
| **Experience with CT for COVID-19 (time)** (%) | |
| • Other CT experience than COVID-19 | 15 (2.3%) |
| • <1 month | 13 (2.0%) |
| • 1–6 months | 137 (21.4%) |
| • 7–12 months | 92 (14.4%) |
| • 1–2 years | 384 (59.9%) |
| **Experience with CT for communicable diseases (other than COVID-19)** (%) | |
| • No | 588 (91.7%) |
| • Yes | 53 (8.3%) |

was nurse or physician and 83.9% was employed as a (temporary) contact tracer in an executive function. Most respondents had 1–2 years of experience with CT for COVID-19 (59.9%) and did not have experience with CT for communicable diseases other than COVID-19 (91.7%). Out of all respondents, 2.3% only had experience with CT for diseases other than COVID-19. See Table 1 for an overview of the sample characteristics.

## Descriptive analyses

Key-findings are presented in the sections below. A more detailed overview of the descriptive analyses can be found in S3 Appendix.

**Intention to use DCTS-tools.** Overall, respondents had a positive intention towards using DCTS-tools 1 (M = 2.73; SD = 0.98), 2 (M = 2.56; SD = 0.96), and 3 (M = 2.51; SD = 1.08). For our multivariable analyses, most respondents were classified as having a positive intention towards using DCTS-tools 1, 2, and 3 (64.5%, 58.0%, and 55.2%, respectively) in CT for COVID-19. Notably, of the respondents classified as having a neutral/negative attitude, a relatively small percentage had a (very) negative intention to use DCTS-tools 1, 2, and 3 (11.5%, 14.1%, and 19.2%, respectively).

**General beliefs regarding CT for COVID-19.** Respondents generally considered CT for COVID-19 a relatively easy (M = 2.7; SD = 0.77) and important (M = 2.39; SD = 0.94) task. Respondents did not consider CT particularly much or little work (M = 1.97; SD = 0.89), nor did they consider it very fast or slow (2.16; SD = 0.91). Overall, respondents believed that CT was more effective for gaining overview and insight into transmission of COVID-19 (2.47; SD = 1.14) than for stopping transmission of COVID-19 (1.94; SD = 1.10).

**Beliefs regarding the roles of cases, contacts, and PHS in CT.** Respondents generally felt that both cases (M = 3.02; SD = 0.68) and contacts (M = 2.41; SD = 0.87) are willing to participate in CT. Similarly, it was felt that cases (M = 2.69; SD = 0.73) and contacts (M = 2.38; SD = 0.79) are generally compliant with the CT-measures communicated to them. It was believed that cases and contacts know roughly what CT entails (M = 2.23; 0.94) but are not always aware of the role of CT in combating COVID-19 (M = 1.78; SD = 0.96). Overall, CT was slightly more often seen as the responsibility of PHS than as that of cases and contacts (M = 1.72; SD = 0.75), and it was felt that PHS-control is necessary to perform CT adequately (M = 2.94; SD = 0.79).

**Beliefs regarding the digitalization of CT.** Overall, respondents had a positive attitude towards digitalization in CT (M = 2.75; SD = 0.77). It was believed that digitalization can make CT more efficient (M = 2.92; SD = 0.84) and that digitalization is necessary to improve CT in the future (M = 3.05; SD = 0.77). Respondents thought that the CT-process may be digitalized to some extent (M = 2.45; SD = 0.91), but personal contact with cases (M = 3.23; SD = 0.77)– and to a lesser extent also with contacts (M = 2.35; SD = 1.01)–was overall still considered as (very) important by respondents. Most respondents indicated that it would be easy for them to learn how to use new digital tools for CT (M = 2.94; SD = 0.84).

**Beliefs regarding benefits and challenges of DCTS-tools 1, 2, and 3 for CT.** We questioned respondents about potential advantages and challenges of using DCTS-tools 1, 2, and 3 in CT for COVID-19, compared to conducting CT without these tools. Notably, respondents anticipated that DCTS-tools 1, 2, and 3 could increase the speed of CT (M = 2.75; SD = 0.78, M = 2.69; SD = 0.79, and M = 2.73; SD = 0.80, respectively), reduce the workload of CT for PHPs (M = 2.76; SD = 0.82, M = 2.75; SD = 0.88, and M = 2.83; SD = 0.88, respectively), and make CT easier to perform for PHPs (M = 2.65; SD = 0.75, M = 2.59; SD = 0.74, and M = 2.67; SD = 0.81, respectively). Another effect that was frequently expected from all DCTS-tools, was a reduction in PHPs' control over the CT-process (M = 1.71; SD = 0.95, M = 1.6; SD = 0.94, and M = 1.65; SD = 0.96, for DCTS-tools 1, 2, and 3, respectively). Other anticipated effects of DCTS-tools (e.g., regarding the support that PHPs can offer to cases and/or contacts in CT, the willingness and skills of cases and/or contacts to participate in CT, and the compliance of cases and/or contacts with CT-measures) were relatively closer to the scale median '2'. This indicates that, on average, it was not anticipated that there would be a large difference with respect to 'regular' CT-practices in the context of COVID-19.

## Multivariable analyses: identifying determinants of PHPs' intention to use DCTS-tools

Three RF-models were constructed to predict PHPs' intention to use DCTS-tools 1, 2, and 3 in CT for COVID-19, respectively. The SPEC (probability of a correct prediction among

**Table 2. RF-model performance indicators.**

| RF model | Specificity (SPEC) | Sensitivity (SENS) | Probability of misclassification (PMC) | Area under the receiver operating characteristic curve (AUC) |
|---|---|---|---|---|
| DCTS-tool 1 | 0.65 | 0.90 | 0.19 | 0.86 |
| DCTS-tool 2 | 0.74 | 0.85 | 0.20 | 0.88 |
| DCTS-tool 3 | 0.79 | 0.83 | 0.19 | 0.88 |

individuals with a neutral/negative intention) was highest in the DCTS-tool 3 model (0.79) and lowest in the DCTS-tool 1 model (0.65). SENS (the probability of a correct prediction among individuals with a positive intention) was higher in the DCTS-tool 1 model (0.90) than in the models for DCTS-tools 2 and 3 (0.85 and 0.83, respectively). The PMC (probability of misclassification) was 0.19 in the models for DCTS-tools 1 and 3, and 0.20 in the DCTS-tool 2 model. The AUC was 0.86 in the DCTS-tool 1 model and 0.88 in the models for DCTS-tools 2 and 3 (See Figs H, I, and J in S3 Appendix). See Table 2 for an overview of the performance indicators.

**Determinants of PHPs' intention to use DCTS-tool 1 in CT for COVID-19.** Based on the VIR of model 1, we identified 16 determinants (i.e., relatively important predictors) of PHPs' intention to use DCTS-tool 1 in the contact identification stage of the CT-process. See Fig 4 and Table 3. The degree to which DCTS-tool 1 was anticipated to impact the speed of CT was the strongest determinant of PHPs' intention. The MP difference for this determinant was 27.8% and the direction of the relationship was positive. This means that, on average, expecting that CT can be performed much faster with DCTS-tool 1 increases the probability of a

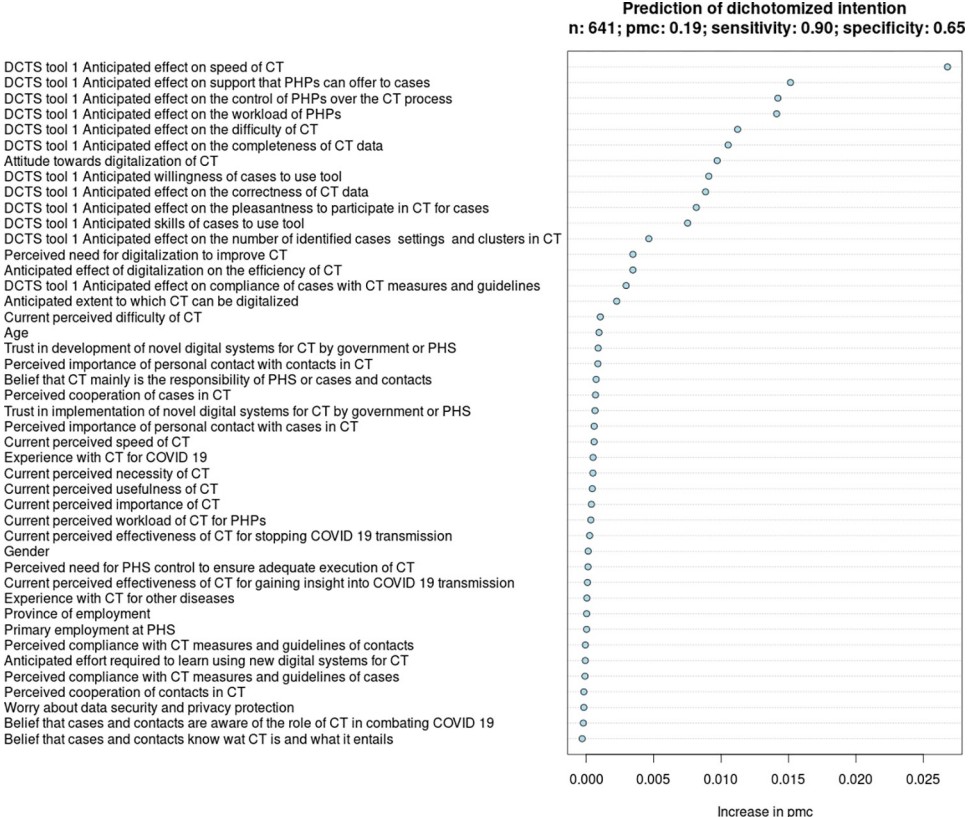

**Fig 4. Variable importance ranking in relation to the intention of PHPs to use DCTS-tool 1.**

**Table 3. Marginal probabilities of determinants of PHPs' intention to use DCTS-tool 1.**

| Determinants in RF-model 1[a], in descending order of importance (Fig 4) | Marginal probabilities, lowest–highest | Change (%) in marginal probability | Direction of relationship |
|---|---|---|---|
| DCTS-tool 1: Anticipated effect on speed of CT | 0.54–0.69 | 27.8% | + |
| DCTS-tool 1: Anticipated effect on support that PHPs can offer to cases | 0.59–0.70 | 18.6% | + |
| DCTS-tool 1: Anticipated effect on the control of PHP over the CT-process | 0.59–0.70 | 18.6% | + |
| DCTS-tool 1: Anticipated effect on the workload of PHPs in CT | 0.57–0.68 | 19.3% | + |
| DCTS-tool 1: Anticipated effect on the difficulty of CT | 0.60–0.68 | 13.3% | + |
| DCTS-tool 1: Anticipated effect on the completeness of data collected in CT | 0.60–0.69 | 15.0% | + |
| Attitude towards digitalization of CT | 0.60–0.68 | 13.3% | + |
| DCTS-tool 1: Anticipated willingness of cases to participate in CT with tool | 0.62–0.68 | 9.7% | + |
| DCTS-tool 1: Anticipated effect on the correctness of data collected CT | 0.60–0.69 | 15.0% | + |
| DCTS-tool 1: Anticipated effect on the pleasantness to participate in CT for cases | 0.61–0.68 | 11.5% | + |
| DCTS-tool 1: Anticipated skills of cases to participate in CT with tool | 0.61–0.68 | 11.5% | + |
| DCTS-tool 1: Anticipated effect on the number of identified cases, settings, and clusters in CT | 0.61–0.67 | 9.8% | + |
| Perceived need for digitalization to improve CT | 0.60–0.67 | 11.7% | + |
| Anticipated effect of digitalization on efficiency of CT | 0.60–0.67 | 11.7% | + |
| DCTS-tool 1: Anticipated effect on compliance of cases with CT-measures and guidelines | 0.63–0.67 | 6.3% | + |
| Anticipated extent to which CT can be digitalized | 0.64–0.66 | 3.1% | + |

[a]See Table B in S3 Appendix for a detailed description of the predictors.

positive intention by 27.8%. Other important determinants included the belief that cases can sufficiently be supported in identifying their contacts and collecting their (health) data when using DCTS-tool 1 (MP difference = 18.6%; direction = +), the anticipated degree to which DCTS-tool 1 would impact the control of PHPs over the CT-process (MP difference = 18.6%; direction = +), and the expected impact of DCTS-tool 1 on the workload of CT for PHPs (MP difference = 19.3%; direction = +).

**Determinants of PHPs' intention to use DCTS-tool 2 in CT for COVID-19.** Based on the VIR of model 2, we identified 16 determinants of PHPs' intention to use DCTS-tool 2 in the contact notification stage of the CT-process. See Fig 5 and Table 4. The strongest determinant of PHPs' intention to use DCTS-tool 2 was the anticipated willingness of cases to use DCTS-tool 2 in CT for COVID-19 (MP difference = 22.6%; direction = +). Other important determinants were the anticipated degree to which DCTS-tool 2 may enhance the speed of CT (MP difference = 24%; direction = +), the perception that cases can sufficiently be supported in notifying their contacts with DCTS-tool 2 (MP difference = 20.4%; direction = +), and the perception that it is (more) pleasant for contacts to be notified through their peers through DCTS-tool 2, compared to being notified by a PHP (MP difference = 23.5%; direction = +).

**Determinants of PHPs' intention to use DCTS-tool 3 in CT for COVID-19.** Based on the VIR of model 3, we identified 12 determinants of PHPs' intention to use DCTS-tool 3 in the contact monitoring stage of the CT-process. See Fig 6 and Table 5.

The strongest determinant of PHPs' intention to use DCTS-tool 3 is the anticipated willingness of contacts to use DCTS-tool 3 for the monitoring and reporting of symptoms (MP difference = 53.3%; direction = +), followed by the perception that it may be more pleasant for contacts to participate in CT with DCTS-tool 3, compared to PHPs performing the monitoring

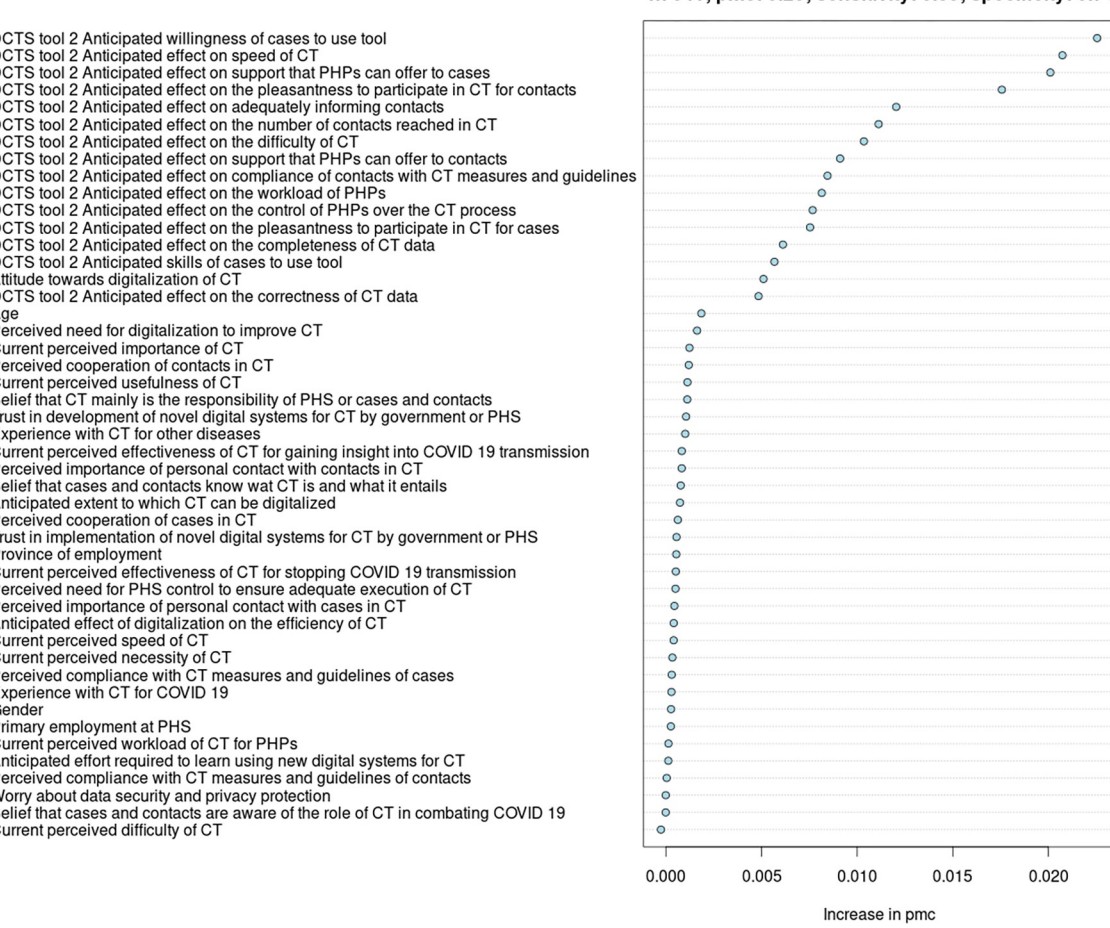

**Fig 5. Variable importance ranking in relation to the intention of PHPs to use DCTS-tool 2.**

(MP difference = 30.6%; direction = +). The degree to which DCTS-tool 3 was anticipated to impact the difficulty of CT for PHPs (MP difference = 25%; direction = +) and the speed of CT (MP difference = 18%; direction = +) also were relatively strong determinants.

**Overall similarities and differences between determinants for PHPs' intention to use DCTS-tools 1, 2, and 3.** From the VIRs (Figs 4–6) and the MPRs (Tables 3–5), it can be noted that PHPs' intention to use DCTS-tools 1, 2 and 3 is dependent on numerous determinants, rather than a few top-determinants. Typically, determinants related to the anticipated impact of DCTS-tools on the feasibility and efficiency of CT (speed, workload, difficulty), support for cases and/or contacts, and the degree to which cases and/or contacts may find it pleasant and may be willing to participate in CT using DCTS-tools, ranked relatively high. Questionnaire items from sections 1–4 (i.e., demographic characteristics, beliefs regarding 'regular' CT-practices, beliefs regarding the roles of cases, contacts and PHS in CT, and beliefs regarding the digitalization of CT, respectively) ranked relatively low.

Nevertheless, several differences stand out. Compared to DCTS-tools 2 and 3, PHPs' general attitude towards digitalization in CT was a relatively strong determinants of their intention to use DCTS-tool 1, whereas the anticipated willingness of cases to use DCTS-tool 1 and the anticipated pleasantness of DCTS-tool 1 for cases were relatively weaker determinants. For

**Table 4. Marginal probabilities of determinants of PHPs' intention to use DCTS-tool 2.**

| Determinants in RF-model 2[a], in descending order of importance (Fig 5) | Marginal probabilities, lowest–highest | Change (%) in marginal probability | Direction of relationship |
|---|---|---|---|
| DCTS-tool 2: Anticipated willingness of cases to participate in CT with tool | 0.53–0.65 | 22.6% | + |
| DCTS-tool 2: Anticipated effect on speed of CT | 0.50–0.62 | 24% | + |
| DCTS-tool 2: Anticipated effect on the support that PHPs can offer to cases | 0.54–0.65 | 20.4% | + |
| DCTS-tool 2: Anticipated effect on the pleasantness to participate in CT for contacts | 0.51–0.63 | 23.5% | + |
| DCTS-tool 2: Anticipated effect on adequately informing contacts | 0.54–0.63 | 16.7% | + |
| DCTS-tool 2: Anticipated effect on the number of contacts reached in CT | 0.56–0.63 | 12.5% | + |
| DCTS-tool 2: Anticipated effect on the difficulty of CT | 0.52–0.61 | 17.3% | + |
| DCTS-tool 2: Anticipated effect on the support that PHPs can offer to contacts | 0.56–0.63 | 12.5% | + |
| DCTS-tool 2: Anticipated effect on compliance of contacts with CT-measures and guidelines | 0.54–0.61 | 13.0% | + |
| DCTS-tool 2: Anticipated effect on the workload of PHPs in CT | 0.52–0.60 | 15.4% | + |
| DCTS-tool 2: Anticipated effect on the control of PHPs over the CT-process | 0.55–0.62 | 12.7% | + |
| DCTS-tool 2: Anticipated effect on the pleasantness to participate in CT for cases | 0.55–0.60 | 9.1% | + |
| DCTS-tool 2: Anticipated effect on the completeness of data collected in CT | 0.57–0.61 | 7.0% | + |
| DCTS-tool 2: Anticipated skills of cases to participate in CT with tool | 0.56–0.61 | 8.9% | + |
| Attitude towards digitalization of CT | 0.53–0.61 | 15.1% | + |
| DCTS-tool 2: Anticipated effect on the correctness of data collected in CT | 0.57–0.61 | 7.0% | + |

[a]See Table B in S3 Appendix for a detailed description of the predictors.

DCTS-tool 2, the number of contacts that can be reached in CT and the degree to which contacts can be adequately informed by their peers were unique and relatively important determinants. Contrastingly, the degree to which DCTS-tool 2 was anticipated to impact the difficulty and workload of CT for PHPs were comparatively less important. For DCTS-tool 3, the anticipated impact on the compliance with CT-measures was a relatively strong determinant, whereas the anticipated impact of DCTS-tool 3 on PHPs' workload was a relatively weaker.

## Discussion

Most PHPs have a positive intention towards using DCTS-tools to involve cases and their contacts in all stages of the CT-process (i.e., contact identification, notification, and monitoring) for COVID-19. Although we observed some differences in the prioritization of individual determinants between the DCTS-tools, the overall groups of variables that determines PHPs' intention to use them appears very similar. In general, important determinants of a positive intention among PHPs are the anticipated effect of DCTS-tools on the feasibility and efficiency of CT (speed, workload, difficulty), the degree to which PHPs anticipated that cases and contacts may find it pleasant and may be willing to participate in CT using DCTS-tools, and the degree to which PHPs anticipated that they could sufficiently support cases and contacts when using DCTS-tools. Addressing these determinants should be prioritized in the future development and implementation of DCTS-tools.

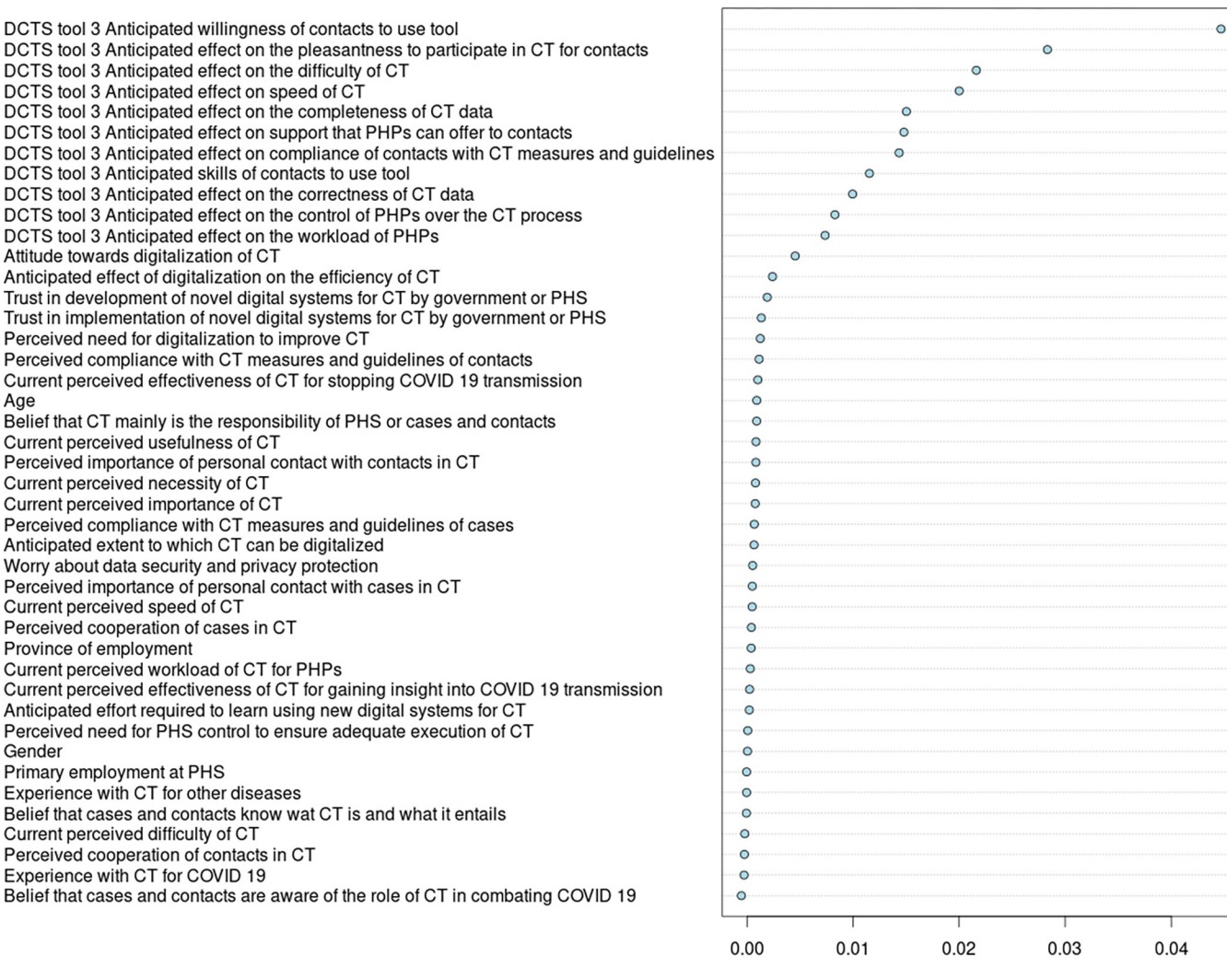

**Fig 6. Variable importance ranking in relation to the intention of PHPs to use DCTS-tool 3.**

## Comparison of findings with literature

Research regarding DCTS-tools is scarce, particularly in the context of close-contact pathogens and–even more so–focusing on the perspectives of PHPs. Nevertheless, some studies that more generally investigated PHPs' perspectives on (digital) contact tracing in the context of COVID-19 yielded similar results. For example, timeliness (i.e., speed) was typically considered the most important attribute of CT-systems by contact tracers across the globe [31]. Another study conducted in New Zealand found that the workload for PHPs and (personal) support for citizens were identified by PHPs as important concerns regarding the use of digital CT-applications [32].

Some studies investigated healthcare professionals' perspectives regarding digital partner notification in the context of STI's. For example, most clinicians in an Australian study about digital partner notification for Chlamydia had a positive attitude towards such a website. Being

**Table 5. Marginal probabilities of determinants of PHPs' intention to use DCTS-tool 3.**

| Determinants in RF-model 1[a], in descending order of importance (Fig 6) | MP, lowest–highest | Change (%) in marginal probability | Direction of relationship |
|---|---|---|---|
| DCTS-tool 3: Anticipated willingness of contacts to participate in CT with tool | 0.45–0.69 | 53.3% | + |
| DCTS-tool 3: Anticipated effect on the pleasantness to participate in CT for contacts | 0.49–0.64 | 30.6% | + |
| DCTS-tool 3: Anticipated effect on the difficulty of CT | 0.48–0.60 | 25.0% | + |
| DCTS-tool 3: Anticipated effect on speed of CT | 0.50–0.59 | 18.0% | + |
| DCTS-tool 3: Anticipated effect on the completeness of data collected in CT | 0.50–0.60 | 20.0% | + |
| DCTS-tool 3: Anticipated effect on the support that PHPs can offer to contacts | 0.48–0.60 | 25.0% | + |
| DCTS-tool 3: Anticipated effect on compliance of contacts with CT-measures and guidelines | 0.50–0.60 | 20.0% | + |
| DCTS-tool 3: Anticipated skills of contacts to participate in CT with tool | 0.51–0.59 | 15.7% | + |
| DCTS-tool 3: Anticipated effect on the correctness of data collected in CT | 0.51–0.58 | 13.7% | + |
| DCTS-tool 3: Anticipated effect on the control of PHPs over the CT-process | 0.53–0.60 | 13.2% | + |
| DCTS-tool 3: Anticipated effect on the workload of PHPs in CT | 0.52–0.57 | 9.6% | + |
| Attitude towards digitalization of CT | 0.51–0.57 | 11.8% | + |

[a]See Table B in S3 Appendix for a detailed description of the predictors.

able to properly support cases in notifying their contacts was an important condition to use digital partner notification and losing a sense of control over the notification process was a barrier to digital partner notification [33]. Healthcare providers in Chile considered online partner notification services a suitable strategy to complement regular partner notification, since it was anticipated that more contacts may be notified through online partner notification. On the other hand, there were concerns that contacts may not be adequately notified about their exposure and informed about CT-measures [34].

The similarities between these studies and the results from our study indicate that PHPs' may have similar perspectives and needs regarding the use of DCTS-tools across different countries and in the context of different communicable diseases. For example, (personal) support for cases and contacts, and considering their preferences, willingness, and skills in the application of digital CT-tools is a recurring theme that consistently seems to lead PHPs to consider DCTS-tools as an addition to—rather than a replacement of—the 'traditional' CT-approach.

In contrast to the above discussed similarities, some widely discussed privacy and data security related concerns in the context of digital CT-applications did not emerge as important determinants of PHPs' intention in this study [35]. Although this was only a minor part of our questionnaire (assessed by one general question in questionnaire section 4), we still see this an unexpected result, also considering the emphasis that was placed on this topic by PHPs in previous qualitative research that we conducted [19,20]. As such, we strongly believe that privacy and data security should remain an important part of the (future) development of DCTS-tools, also because citizens (on whom the success of DCTS-tools ultimately depends) may value these issues more strongly [36,37] and because it is legally required to ensure certain standards when digitally collecting, exchanging, and storing medical and sensitive personal information [38].

In this study, we based the development of questionnaire items on previous research in which we used the Innovation Decision-Process model and the Reasoned Action Approach as theories to elicit PHPs' beliefs that influenced their intention to use DCTS-tools in CT. Given that our RF-models can predict PHPs' intention quite well (an AUC of 0.8–0.89 can be seen as

'good' [25]), we may conclude that the initially chosen theories have a good fit with the study subject and objectives. Nevertheless, we would like to note that our results also seem to align well with more frequently used theories to investigate healthcare professionals' acceptance or use of digital/telecommunications technologies to deliver health care, such as the Unified Theory of Acceptance and Use of Technology (UTAUT) and the Technology Acceptance Model (TAM) [39]. For example, the most important determinants identified in this study seem closely related to perceived usefulness/performance expectancy (e.g., anticipated impact of DCTS-tools on speed and workload of CT), perceived ease of use/effort expectancy (e.g., anticipated impact of DCTS-tools on the difficulty of CT), and social influence (e.g., anticipated willingness of cases/contacts to use DCTS-tools in CT), all of which are important constructs of the TAM and the UTAUT. In addition, it has been demonstrated that constructs from the Health Believe Model (HBM), such as perceived severity, may further improve the performance of technology acceptance models in explaining the adoption of digital CT-applications [40]. As such, we believe that more explicitly taking technology acceptance models (TAM and/or UTAUT) into account, with potential additions from the HBM, may further enhance our understanding of PHPs' intention to use DCTS-tools in future research. Such a multi-theory approach also fits the problem-drive nature of our research [41].

## Strengths and limitations

An important strength of our study is that we managed to collect data from a diverse sample of PHPs, who were directly involved in the execution of CT during the COVID-19 pandemic, at a point in time at which CT was executed at a very large scale and PHS' resources were under significant pressure. Hence, these data are unique and highly valuable for future (large scale) outbreaks.

Another important strength of this study is that we developed our online questionnaire based on previous exploratory work, meaning that questionnaire items covered a wide range of subjects particularly relevant to the study subject and context. We believe that this is also reflected in the performance of the RF-models, all of which had good AUC-values.

At the same time, we are aware that the results from this study are likely strongly influenced by the circumstances under which CT was performed at the time that this study was conducted. For example, most PHS in the Netherlands had insufficient human resources available (considering the number of daily confirmed new cases) to thoroughly conduct CT. As such, making the CT-process more feasible and efficient was an important task for PHS, which we believe to be reflected in the fact that related items (speed, workload, difficulty) ranked relatively highly in the VIR of all DCTS-tools. It may well be that the prioritization of items will be different under different circumstances, both in terms of the stage of an outbreak (e.g., a new or an established outbreak) and in terms of the specific disease at hand. However, considering the similarities in results between this study, previous exploratory research that we conducted before the COVID-19 pandemic on the same topic, and several other studies (as previously outlined), we still strongly believe that our results and recommendations are relevant regarding the application of DCTS-tools beyond the COVID-19 pandemic.

Since we mainly relied on referrals and snowball sampling to distribute the online questionnaire, we were unable to calculate a response rate. In addition, we have no precise insights into characteristics of the overall population of PHPs at the time this study was conducted. This means that we cannot accurately estimate to what degree our study may have suffered from selection bias. Furthermore, our sample consisted mainly of temporary contact tracers, which may not represent the overall population of PHPs under 'normal' (i.e., non-pandemic) circumstances. This should be kept in mind when interpreting this study's results and findings.

Nevertheless, we would like to emphasize that we managed to collect data from a large number of PHPs (N = 641), representing all relevant socio-demographic and professional backgrounds as described in Table 1. Even though our sample consisted largely of temporary contact tracers, our RF-models show that PHPs' socio-demographic and professional characteristics had a negligible effect on their intention to use DCTS-tools. This indicates that potential bias in our study related to over- or underrepresentation of professional backgrounds may be limited. As previously described, we also established that the potential bias in our study related to respondent dropout is limited. Therefore, we argue that we have obtained a good overall picture of PHPs' attitudes and the factors influencing their intention to use DCTS-tools in the Netherlands, especially under pandemic circumstances and/or when large-scale CT is required.

### Recommendations

We identified many determinants of PHPs' intention (16, 16, and 12, for DCTS-tools 1, 2, and 3, respectively), suggesting a complex determination of PHPs' intention to use DCTS-tools in CT for COVID-19. In addition, many determinants correlate (moderately to strongly) amongst each other (see also Figs A, B, and C in S3 Appendix). We believe that this is largely because questionnaire items were derived from qualitatively identified overarching themes, meaning that items that were derived from the same or related (sub-) themes are conceptually related or overlapping. This overlap is also grounded in theories on behavior [27]. For the future development and implementation of DCTS-tools in CT for COVID-19 (and epidemiologically similar pathogens or outbreak scenario's) we, therefore, believe that our results only partly justify prioritizing individual items over others, especially regarding those derived from the same qualitative (sub-)themes and/or those correlating relatively strongly amongst each other. Rather, we suggest focusing on groups, or 'clusters' of highly ranked *and* related determinants in conjunction.

Based on our results we believe that there are three clusters of determinants that may be prioritized in the development and implementation of DCTS-tools for COVID-19. For more information on how we established/identified the clusters we refer to Figs D, E, and F in S3 Appendix, where we show the output of agglomerative hierarchical cluster analyses that we performed as part of our (multi)collinearity assessment. The first cluster, *'enhancing the feasibility and efficiency of CT'*, includes the workload, speed, and difficulty of CT. The second cluster, *'sufficiently supporting and overseeing cases and contacts to ensure adequate execution of CT'*, includes support for cases/contacts and control of PHPs over the CT-process, and variables related to the performance of CT, such as the correctness and completeness of CT-data and the number of contacts identified/notified. The third cluster, *'considering the preferences, willingness, and skills of cases and contacts in the application of DCTS-tools'*, includes variables related to the pleasantness of CT for cases/contacts, and their willingness and skills to participate in CT with DCTS-tools.

In line with the identified clusters, we provide some elements to be considered for further implementation in practice. Note that these suggestions are based on our own personal insights and knowledge of behavior change methods [42], and insights from previous research that we conducted [19,20]. To enhance the feasibility and efficiency of CT, we suggest that DCTS-tools could facilitate automatic data transfer to the PHS' case management system. In addition, cases and contacts could be asked to use DCTS-tools as soon as possible in the CT-process, ideally already before they are contacted by a PHP (e.g., when tested or upon receiving the positive test result). To sufficiently support and oversee cases and contacts when participating in CT with DCTS-tools, we suggest that guidelines and instructions may be provided in

the DCTS-tools and DCTS-tools could provide opportunities for cases/contacts to communicate with PHPs (e.g., through a chat-functionality). To consider the preferences, willingness, and skills of cases and contacts in the application of DCTS-tools, we suggest that DCTS-tools may not be provided as the only option for CT, so that 'regular' CT, guided by a PHP, remains available to cases and contacts.

On a final note, we would like to stress that the 'success' of DCTS-tools ultimately depends on their actual uptake. Therefore, citizens' perspectives and needs regarding DCTS-tools should similarly be considered in the development and implementation of DCTS-tools. To this purpose, we conducted several studies among citizens (manuscripts in preparation), based on which we aim to identify potential synergies or areas of conflict with PHPs' perspectives to further guide our recommendations regarding the implementation of DCTS-tools in practice.

## Conclusions

Most PHPs have a (very) positive intention to involve cases and their contacts in the identification, notification, and monitoring stages of the CT-process through DCTS-tools. In general, enhancing the feasibility and efficiency of CT, considering the willingness of cases and contacts to use DCTS-tools, and support for cases and contacts were the most important determinants of PHPs' intention to use DCTS-tools. These determinants should be prioritized in their (future) development and implementation. Citizens' perspectives should similarly be investigated and considered in the future.

## Supporting information

**S1 Appendix. COREQ-checklist.**
(DOC)

**S2 Appendix. Questionnaire (translated from Dutch to English).**
(DOCX)

**S3 Appendix. Supplementary statistical analyses.**
(DOCX)

**S1 Dataset. Minimal dataset used to produce the results presented in this study.** In this dataset, we aggregated the variables 'Age' and 'Province of employment' to increase the anonymity of respondents.
(CSV)

## Author Contributions

**Conceptualization:** Yannick B. Helms, Mart L. Stein, Nora Hamdiui, Aura Timen, Mirjam E. E. Kretzschmar.

**Formal analysis:** Yannick B. Helms, José A. Ferreira.

**Funding acquisition:** Mart L. Stein, Mirjam E. E. Kretzschmar.

**Investigation:** Yannick B. Helms.

**Methodology:** Yannick B. Helms, Mart L. Stein, Nora Hamdiui, José A. Ferreira, Rik Crutzen, Aura Timen, Mirjam E. E. Kretzschmar.

**Project administration:** Mart L. Stein, Nora Hamdiui.

**Supervision:** Mart L. Stein, Nora Hamdiui, Aura Timen, Mirjam E. E. Kretzschmar.

**Visualization:** Yannick B. Helms, José A. Ferreira.

**Writing – original draft:** Yannick B. Helms.

**Writing – review & editing:** Yannick B. Helms, Mart L. Stein, Nora Hamdiui, Akke van der Meer, José A. Ferreira, Rik Crutzen, Aura Timen, Mirjam E. E. Kretzschmar.

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
