## [Decision Letter · Decision Letter 0]

3 Oct 2023

PDIG-D-23-00330

Determinants of Dutch public health professionals’ intention to use digital contact tracing support tools: a cross-sectional online questionnaire study

PLOS Digital Health

Dear Dr. Helms,

Thank you for submitting your manuscript to PLOS Digital Health. After careful consideration, we feel that it has merit but does not fully meet PLOS Digital Health's publication criteria as it currently stands. Therefore, we invite you to submit a revised version of the manuscript that addresses the points raised during the review process.

Please submit your revised manuscript within 60 days Dec 02 2023 11:59PM. If you will need more time than this to complete your revisions, please reply to this message or contact the journal office at digitalhealth@plos.org. Please include the following items when submitting your revised manuscript:

We look forward to receiving your revised manuscript.

Kind regards,

Haleh Ayatollahi

Section Editor

PLOS Digital Health

Journal Requirements:

Additional Editor Comments (if provided):

Reviewers' comments:

Reviewer's Responses to Questions

**Comments to the Author**

1. Does this manuscript meet PLOS Digital Health’s publication criteria? Is the manuscript technically sound, and do the data support the conclusions? The manuscript must describe methodologically and ethically rigorous research with conclusions that are appropriately drawn based on the data presented.

Reviewer #1: Yes

Reviewer #2: Yes

Reviewer #3: Yes

2. Has the statistical analysis been performed appropriately and rigorously?

Reviewer #1: Yes

Reviewer #2: Yes

Reviewer #3: Yes

3. Have the authors made all data underlying the findings in their manuscript fully available (please refer to the Data Availability Statement at the start of the manuscript PDF file)?

Reviewer #1: No

Reviewer #2: Yes

Reviewer #3: Yes

4. Is the manuscript presented in an intelligible fashion and written in standard English?

Reviewer #1: Yes

Reviewer #2: Yes

Reviewer #3: Yes

5. Review Comments to the Author

Reviewer #1: I welcome the opportunity to review this manuscript. Here are some suggestions for improvements.

Introduction

Regarding your previous research, Can you add the sample size of your previous research to support your justification for this further research with additional survey size .

I suggest the authors identify and present their research objectives, hypothesis and the question / questions that your survey was designed to answer. The research questions should then flow through the results and into the discussion sections. 

I suggest the authors expand the introduction to include more information about the study setting(s) - to more effectively set the scene for the methods section.

Methods 

The authors might want to consider describing any efforts to address potential sources of bias and whether any sources of bias should be mentioned as a limitation.

Were any survey responses incomplete and was any data cleansing performed?

Results

As most of the respondents had only relatively little experience in CT - being temp workforce and no other experience with other diseases, do the authors feel this is a limitation that warrants including in the relevant section?

Discussion 

The authors might want to consider whether reflecting on any differences between the results of the 3 tools would add value within the discussion

The authors offer some suggestions for functionality based on the factors influencing intent to use DHTs of the respondents. Authors should make this clear how these recommendations were identified and whether this is the opinion of the authors or whether these recommendations have been tested or informed by other research.

I suggest making the predictors of intent that were grouped together to be made clearer.

Reviewer #2: . Yes, this s manuscript meet PLOS Digital Health's publication criteria as evidenced by the sound methodology, ethically rigorous research with realistic conclusion in study: - see lines 57 -565)

2. Yes, supported by the methodology (lines 226 - 429)

3. Yes, The authors made all data underlying the findings in their manuscript as per statement at the start of the manuscript file.

4. Yes, the manuscript presented in standard English- readable, great syntax, clear and organized

Reviewer #3: I read the study with interest and found it to be valuable. However, I noticed that the authors did not provide a clear explanation of the generalization procedure, which should be addressed.

The study was conducted between February and April 2022, during a non-pandemic situation. However, the authors did not mention why they chose online questionnaires instead of face-to-face interviews.

Data were collected from the PHPs involved in CT for COVID-19, which could have resulted in recall bias. Unfortunately, the authors did not address this issue in the limitations section.

The study included a large number of temporary contact tracers (538 or 83.9%) and some participants involved as young as 16 years old. Thus, the sample group may not accurately reflect actual public health professionals.

To make the study more reliable, a number of Key Informant Interviews should have been conducted.

6. PLOS authors have the option to publish the peer review history of their article (what does this mean?). If published, this will include your full peer review and any attached files.

**Do you want your identity to be public for this peer review?** For information about this choice, including consent withdrawal, please see our Privacy Policy.

Reviewer #1: Yes: Shoshana Bloom

Reviewer #2: Yes: Dr. Audrey Blackwood, RHIA, LPC-S

Reviewer #3: Yes: Towhida Ahsan

---

## [Decision Letter · Decision Letter 1]

17 Nov 2023

PDIG-D-23-00330R1

Determinants of Dutch public health professionals’ intention to use digital contact tracing support tools: a cross-sectional online questionnaire study

PLOS Digital Health

Dear Dr. Helms,

Thank you for submitting your manuscript to PLOS Digital Health. After careful consideration, we feel that it has merit but does not fully meet PLOS Digital Health's publication criteria as it currently stands. Therefore, we invite you to submit a revised version of the manuscript that addresses the points raised during the review process.

Please submit your revised manuscript within 30 days Dec 17 2023 11:59PM. If you will need more time than this to complete your revisions, please reply to this message or contact the journal office at digitalhealth@plos.org. Please include the following items when submitting your revised manuscript:

We look forward to receiving your revised manuscript.

Kind regards,

Haleh Ayatollahi

Section Editor

PLOS Digital Health

Journal Requirements:

Additional Editor Comments (if provided):

Thank you very much for your time and efforts to revise the manuscript. I appreciate if you please apply some further minor revisions to your manuscript to improve it:

1- Please follow the journal instructions for preparing the manuscript, e.g. Unstructured/structured abstract and appropriate heaadings/subheadingss.

2- Please add appropriate keywords using the MeSH terms.

3- Please add a reference to the 1st paragraph of the introduction.

4- Please ensure that the aim of the study has been clearly mentioned in the abstarct and introduction.

Reviewers' comments:

Reviewer's Responses to Questions

**Comments to the Author**

1. If the authors have adequately addressed your comments raised in a previous round of review and you feel that this manuscript is now acceptable for publication, you may indicate that here to bypass the “Comments to the Author” section, enter your conflict of interest statement in the “Confidential to Editor” section, and submit your "Accept" recommendation.

Reviewer #1: All comments have been addressed

Reviewer #3: (No Response)

2. Does this manuscript meet PLOS Digital Health’s publication criteria? Is the manuscript technically sound, and do the data support the conclusions? The manuscript must describe methodologically and ethically rigorous research with conclusions that are appropriately drawn based on the data presented.

Reviewer #1: Yes

Reviewer #3: (No Response)

3. Has the statistical analysis been performed appropriately and rigorously?

Reviewer #1: Yes

Reviewer #3: (No Response)

4. Have the authors made all data underlying the findings in their manuscript fully available (please refer to the Data Availability Statement at the start of the manuscript PDF file)?

Reviewer #1: No

Reviewer #3: Yes

5. Is the manuscript presented in an intelligible fashion and written in standard English?

Reviewer #1: Yes

Reviewer #3: Yes

6. Review Comments to the Author

Reviewer #1: I am content that all my previous comments have been addressed satisfactorily in this revised manuscript. Many thanks for the opportunity to review this work.

Reviewer #3: We would like to thank the authors for their contributions and responses to the comments.

7. PLOS authors have the option to publish the peer review history of their article (what does this mean?). If published, this will include your full peer review and any attached files.

**Do you want your identity to be public for this peer review?** For information about this choice, including consent withdrawal, please see our Privacy Policy. 

Reviewer #1: Yes: Shoshana Bloom

Reviewer #3: Yes: 

---

## [Editor Report · Decision Letter 2]

2 Dec 2023

Determinants of Dutch public health professionals’ intention to use digital contact tracing support tools: a cross-sectional online questionnaire study

PDIG-D-23-00330R2

Dear Helms,

We are pleased to inform you that your manuscript 'Determinants of Dutch public health professionals’ intention to use digital contact tracing support tools: a cross-sectional online questionnaire study' has been provisionally accepted for publication in PLOS Digital Health.

Best regards,

Haleh Ayatollahi

Section Editor

PLOS Digital Health